# Enhancing Solubility and Dissolution Rate of Antifungal Drug Ketoconazole through Crystal Engineering

**DOI:** 10.3390/ph16101349

**Published:** 2023-09-25

**Authors:** Hongmei Yu, Li Zhang, Meiju Liu, Dezhi Yang, Guorong He, Baoxi Zhang, Ningbo Gong, Yang Lu, Guanhua Du

**Affiliations:** 1Beijing Key Laboratory of Polymorphic Drugs, Institute of Materia Medica, Chinese Academy of Medical Sciences, Peking Union Medical College, Beijing 100050, China; 18612429998@163.com (H.Y.); zhangl@imm.ac.cn (L.Z.); major199810@163.com (M.L.); ydz@imm.ac.cn (D.Y.); zhangbx@imm.ac.cn (B.Z.); 2Beijing City Key Laboratory of Drug Target Identification and Drug Screening, Institute of Materia Medica, Chinese Academy of Medical Sciences, Peking Union Medical College, Beijing 100050, China; hegr@imm.ac.cn (G.H.); dugh@imm.ac.cn (G.D.)

**Keywords:** cocrystal, salt, ketoconazole, solubility, dissolution

## Abstract

To improve the solubility and dissolution rate of the BCS class II drug ketoconazole, five novel solid forms in 1:1 stoichiometry were obtained upon liquid-assisted grinding, slurry, and slow evaporation methods in the presence of coformers, namely, glutaric, vanillic, 2,6-dihydroxybenzoic, protocatechuic, and 3,5-dinitrobenzoic acids. Single-crystal X-ray diffraction analysis revealed that the hydroxyl/carboxylic acid. . .N-imidazole motif acts as the dominant supramolecular interaction in the obtained solid forms. The solubility of ketoconazole in distilled water significantly increased from 1.2 to 2165.6, 321.6, 139.1, 386.3, and 191.7 μg mL^−1^ in the synthesized multi-component forms with glutaric, vanillic, 2,6-dihydroxybenzoic, protocatechuic, and 3,5-dinitrobenzoic acid, respectively. In particular, the cocrystal form with glutaric acid showed an 1800-fold solubility increase in water concerning ketoconazole. Our study provides an alternative approach to improve the solubility and modify the release profile of poorly water-soluble drugs such as ketoconazole.

## 1. Introduction

Ketoconazole (*cis*-1-acetyl-4-[4-[[2-(2,4-dichlorophenyl)-2-(1-*H*-imidazole-1-yl methyl)-1,3-dioxolan-4-yl] methoxy] phenyl] piperazine, KTZ, Figure 1), is a broad-spectrum synthetic imidazole-derived antifungal agent [1,2,3]. It is highly effective against superficial and systematic mycoses [4] administered orally or topically [5]. The action mechanism of KTZ is that it is a potent inhibitor of ergosterol biosynthesis in Candida albicans, which is the main sterol in the membranes of fungi [6,7]. KTZ is a weakly basic drug (p*K*_a_ value of 2.94 and 6.51) [8] and is classified as a Biopharmaceutics Classification Scheme (BCS) class II [9]. Statistics and reports have indicated that KTZ is almost insoluble in water (0.087 mg L^−1^ at 25 °C) and exhibits strong pH-dependent solubility [10]; it is insoluble at higher pH values and sparingly soluble under an acidic medium. This strong pH dependence results in erratic absorption and a broad range of bioavailability, ranging from 37% to 97% [11,12]. Therefore, it is imperative for us to take measures to enhance the solubility and oral bioavailability of KTZ at the same time [13,14,15,16].

Cocrystals and salts are long-known multi-component crystalline assemblies and have been attracting expanding interest among pharmaceutical scientists [17,18,19,20]. Cocrystals and/or salts are solids composed of two or more different molecular and/or ionic compounds in a definite stoichiometric ratio, and the difference is whether there is a proton transfer process [21,22]. Recently, they have emerged as a reliable and sustainable alternative to traditional drug formulation by modifying the physicochemical properties of active pharmaceutical ingredients (APIs), especially the dissolution and solubility properties, without affecting the intrinsic pharmacological activity of APIs [23,24,25,26]. Additionally, a new cocrystal or salt can serve as a contender for a new patent in drug formulation owing to its novelty, efficacy, economic feasibility, and easy synthetic procedure [27]. The supramolecular synthon approach [28,29] is the key to designing and synthesizing new cocrystals and salts. Based on this principle, it is possible to find potential cocrystal coformers (CCFs) that have complementary functionalities to KTZ for the construction of non-covalent interactions.

To date, crystal structures of KTZ salt with oxalate and cocrystals with some organic acids have already been reported [30,31,32,33,34]. The choice of CCFs was based on the following considerations: on the one hand, based on the supramolecular homosynthon and heterosynthon, considering the existence of imidazole and a piperazine ring in the KTZ structure and the high persistence of COOH/OH. . .N synthons in cocrystals between the nitrogen heterocycle complex and dicarboxylic and phenolic acids [14,35,36,37,38], we were encouraged to investigate the possibility of KTZ cocrystallization with dicarboxylic acids and phenolic acids. It is likely to form (C=O)−O−H. . .N heterosynthons when co-crystallizing KTZ with carboxylic acids. On the other hand, GTA, VNA, 26DHB, and PCA were selected from the generally regarded as safe (GRAS) compound list approved by the Food and Drug Administration [39], which is feasible in terms of pharmaceutical development. Furthermore, the solubility of CCFs used in this study is significantly higher than that of KTZ raw material, which has the potential to improve the solubility of KTZ. Five novel multi-components including four cocrystal forms with glutaric acid (GTA), vanillic acid (VNA), protocatechuic acid (PCA), 3,5-dinitrobenzoic acid (35DNB), and one salt form with 2,6-dihydroxybenzoate anion (26DHB), namely, KTZ−GTA, KTZ−VNA, KTZ−PCA, KTZ−35DNB, KTZ−26DHB in 1:1 stoichiometry have been successfully synthesized.

In this study, we mainly focus on crystallographic investigations, including structural analysis, the elucidation of ionization states, and the conformation overlay of four novel crystalline forms as well as stability tests, explicit solubility evaluations concerning equilibrium solubility, and the powder dissolution rate. The five solid forms were also characterized using various solid-state analytical techniques, including powder X-ray diffraction (PXRD), differential scanning calorimetry (DSC), thermogravimetric analysis (TGA), and Fourier transform infrared spectroscopy (FT−IR). Analysis using these complementary characterizations provides a broad range of information on the physicochemical properties at the crystallographic, thermal, and molecular levels.

## 2. Results and Discussion

### 2.1. Crystallographic Results

The crystallographic data and structure refinement details of KTZ−GTA, KTZ−PCA, KTZ−35DNB, and KTZ−26DHB are summarized in Table 1, and ortep diagrams have been provided in Appendix A, and the crystallographic data of KTZ−VNA has been provided in Appendix A due to the poor quality of the crystal structure.

KTZ−GTA (1:1) cocrystal The crystal structure was solved in the monoclinic *P*2_1_/*c* space group (*Z* = 4). The O_3_, C_8_, C_9_, and C_10_ atoms of the KTZ molecule are disordered over two sites with an occupancy ratio of 0.61 (O_3_, C_8_, C_9_, C_10_): 0.39 (O_3A_, C_8A_, C_9A_, C_10A_). The asymmetric unit consists of one KTZ and one GTA molecule (*Z′* = 1, Figure 2a), which is connected via O_8_−H_8_. . .N_4_ interaction between one carboxylic hydroxyl group of GTA and the imidazole N_4_ atom of KTZ, while the carboxylic hydroxyl group on the other side of the GTA molecule connected the carbonyl group of KTZ through O_8_−H_8_. . .O_4_ hydrogen bonds (Figure 2b). In this crystal structure, the KTZ molecules and GTA molecules appear alternately and stack along the crystallographic b-axis through intermolecular hydrogen bonds and other non-covalent contacts to form molecular chains (Figure 2b).

KTZ−26DHB (1:1) salt It crystallizes in the *P*2_1_ space group of the monoclinic system, with two asymmetric units per unit cell (*Z* = 2). Each asymmetric unit contains one KTZ cation and one 26DHB anion (*Z′* = 1, Figure 3a). Proton transfer occurs from the carboxyl acid group of 26DHB to the imidazole N_4_ atom of the KTZ molecule, thus resulting in the formation of a charge-assisted N_4_−H_4_. . .O_6_ ionic interaction. The adjacent two asymmetric units are linked via C_25_−H_25_. . .O_4_ hydrogen bonds formed between the neighboring two KTZ molecules along the crystallographic b-axis, forming a “zig-zag” molecular chain-like structure (Figure 3b,c).

KTZ−PCA-II (1:1) cocrystal It crystallizes in the *P* − 1 space group of the triclinic system, with two asymmetric units per unit cell (*Z* = 2). Each asymmetric unit contains one KTZ molecule and one PCA molecule (*Z*′ = 1, Figure 4a), which are connected via the O_8_−H_8_. . .N_4_ interaction between one acid hydroxyl group and the imidazole N_4_ atom of KTZ, without proton transfer from the acid moiety to the basic moiety (Figure 4a). An intramolecular O_7_−H_7_. . .O_8_ interaction is observed in each PCA molecule. Two adjacent asymmetric units are assembled into an annular tetrameric unit through the participation of O_8_−H_8_. . .N_4_ and O_6_−H_6_. . .O_4_ intermolecular hydrogen bonds (Figure 4b). Two adjacent circuit networks are connected via O_7_−H_7_. . .O_8_ homosynthons and infinitely expanded (Figure 4c). KTZ−PCA−I (1:1) [31] also crystallizes in the *P* − 1 space group of the triclinic system. It is worth noting that the hydroxyl. . .N-imidazole hydrogen bond O_8_−H_8B_. . .N_4_ appears as the main supramolecular heterosynthon in this structure (Appendix A). Along the c-axis, the annular tetramers propagate through the homosynthon O_7_−H_7_. . .O_8_ formed between the two adjacent PCA molecules (Appendix A). Considering the similar hydrogen-bonding interactions and different packing styles between KTZ−PCA−I (1:1) and KTZ−PCA−II (1:1), these two polymorphic forms could be categorized as packing polymorphs.

KTZ−35DNB (1:1) cocrystal The crystal structure belongs to the *P*2_1_/*c* space group of the monoclinic system (*Z* = 4). The asymmetric unit consists of one KTZ molecule and one 35DNB molecule (*Z*′ = 1, Figure 5a), which are connected via the O_6_−H_6_. . .N_4_ hydrogen bond between one carboxylic acid group and the imidazole N_4_ atom of KTZ, without proton transfer (Figure 5a). The O_5_ atom of the 35DNB molecule was found to be disordered over two sites with occupancies of 50% each (for O_5A_ and O_5B_, respectively). Two adjacent asymmetric units are connected via a C_23_−H_23B_. . .O_4_ hydrogen bond between two KTZ molecules (Figure 5b). KTZ molecules connect and expand in a meandering way with 35DNB molecules viewed along the crystallographic c-axis (Figure 5c).

### 2.2. Elucidation of Ionization States

To understand the crystal nature of the obtained solid forms (a salt or a cocrystal), three commonly used principles have been summarized, such as the Δp*K*_a_ rule [40], the position of the hydroxyl-H atom retrieved from SCXRD data, and the structural characteristics of interacting groups [41,42]. The p*K*_a_ values of the starting functional groups, the Δp*K*_a_ values between the two considered groups, the geometric parameters of the C−O bond lengths of the carboxylic/carboxylate group, and the endocyclic C−N−C bond angle are provided in Table 2. Firstly, the Δp*K*_a_ rule has long been used for the prediction of salt or cocrystal formation. It is generally accepted that when Δp*K*_a_ is >4, proton transfer is expected to occur and salt is anticipated to form; when Δp*K*_a_ < −1, on the other hand, the formation of non-ionized cocrystals prevails over that of salts. However, if the Δp*K*_a_ values are located in the region of −1 to 4, it is impossible to provide a prediction of a salt or a cocrystal. The molecular/ionization states of the four complexes could not be fully disclosed from the Δp*K*_a_ values, since the four values fell in the range of −1 to 4, but the most positive value of KTZ−GTA indicates a higher probability of salt formation. Secondly, the residual density map provided by the SXRD data (Appendix A) could be used to distinguish the location of the H atom and, thus, the cocrystal, salt, and salt–cocrystal continuum could be distinguished [43]. In the KTZ−GTA, KTZ−PCA-II, and KTZ−35DNB, the H atom was close to the oxygen atom on the carboxylic hydroxyl group, which could be regarded as cocrystals, while in the KTZ−26DHB, the H atom was close to the imidazole-N atom of the KTZ molecule, suggesting the salt nature of KTZ−26DHB. Thirdly, the ionization state of the structures was also supported by the difference in the C−O bond lengths in a carboxylic/carboxylate group and the bond angle of C_24_−N_4_−C_25_ in the imidazole ring [36,44,45,46]. In a neutral carboxylic group, the length of C−O is longer than that of C=O, while in the carboxylate anion, there is nearly no difference between the bond lengths of C−O and C−O^−^ [47]. The endocyclic C−N−C bond angle of the cationic N-heterocycle is higher than that of the neutral state [48,49]. The cocrystal nature of KTZ−GTA, KTZ−PCA-II, and KTZ−35DNB was supported by the neutral nature of CCF with a longer bond length of C−O than C=O differing by more than 0.05 Å and an approximate endocyclic C_24_−N_4_−C_25_ bond angle concerning the KTZ starting material. The salt nature of KTZ−26DHB was evidenced by the approximate bond length of C=O (1.252 Å) and C−O^−^ (1.280 Å) differing by less than 0.03 Å (0.003 Å) and a significantly increased C_24_−N_4_−C_25_ bond angle compared with KTZ.

### 2.3. Conformation Analysis

In the KTZ molecule of these four multi-components, the two phenyl rings (R2, R4) and the imidazole ring (R5) of the KTZ molecule are planar, while the piperazine ring (R1) and the 1,3-dioxolan ring (R3) adopt a chair and a twisted conformation, respectively. The conformation overlay of KTZ is conducted by superimposing R1 and R2 (Figure 6). The conformation of the KTZ molecule is distinctly different in these four multi-components mainly concerning the orientations of the ring planes. It could be concluded that the introduction of CCF molecules into the crystal lattice or hydrogen bonds between the API and CCF enables the torsion of the API conformation, and, at the same time, the API molecule with flexible conformation forms a more matching spatial arrangement between the API and CCF, which promotes the formation of cocrystals [35,50].

### 2.4. PXRD Analysis

The PXRD patterns of KTZ, the experimental and simulated multi-components, and the five CCFs are illustrated in Figure 7. It can be seen that the featured diffraction peaks of the starting materials cannot be detected in the patterns of the corresponding cocrystal samples. All the experimental patterns are consistent with those calculated from the simulated result of the SCXRD data, indicating that the powder samples possess satisfactory phase purity.

### 2.5. DSC Analysis

The DSC and TGA traces of KTZ and its multi-components are shown in Figure 8, and the DSC thermograms of the CCFs are provided in Appendix A. The melting points of KTZ, GTA, VNA, 26DHB, PCA, and 35DNB are 152.5, 99.4, 211.4, 169.3, 204.0, and 205.9 °C, respectively. The melting points of KTZ−GTA (134.3 °C), KTZ−26DHB (155.9 °C), and KTZ−35DNB (152.5 °C) are in between those of KTZ and the corresponding CCFs, while the melting points for KTZ−VNA (145.9 °C) and KTZ−PCA-II (151.5 °C) are lower than those of the starting components. The melting points of KTZ-26DHB, especially for KTZ-PCA and KTZ-35DNB, are too close to that of API. The individual endothermic peak indicates the high crystalline and phase purity of the obtained crystals, and there is no phase conversion in all of the synthesized solid forms. The TG thermograms suggest solvent-free properties of these five multi-component crystals, and the first weight loss step is accompanied by the decomposition process.

### 2.6. FT−IR Analysis

FT−IR comparisons of pure KTZ and the CCFs with the synthesized multi-component crystals were conducted to distinguish a cocrystal from a salt form, especially the frequency shifts of −C=O stretching vibration, which is characteristic of intermolecular interactions (Table 3), and the results are summarized in Table 3 and Appendix A. There are blue shifts of the wave number of the carbonyl −C=O stretching vibration in the KTZ−GTA, KTZ−VNA, KTZ−PCA-II, and KTZ−35DNB cocrystals, while red shifts were observed to occur in the KTZ−26DHB salt. Shifts of −C=O stretching vibration help to confirm the formation of a KTZ cocrystal or salt [51]. Also, as is labeled in Appendix A, the FT−IR spectra in the 2900−3500 cm^−1^ frequency region (hydroxyl −OH stretching) are not a simple superposition of the corresponding raw materials. These spectral shifts collaboratively reflect that the carbonyl and hydroxyl groups of the CCFs participate in the formation of multi-component crystals.

### 2.7. Equilibrium Solubility

Powder dissolution and equilibrium solubility experiments were both evaluated since the bioavailability and efficacy of a drug largely depend upon its solubility and dissolution rate in an aqueous medium. Equilibrium solubility and powder dissolution experiments perform the solubility assessments from different aspects. Powder dissolution assesses the cocrystal dissolution profile and shows non-equilibrium solubility driven by the cocrystal stability in the solution. Equilibrium solubility refers to the real solubility (thermodynamic solubility) when all the components achieve entire dynamic equilibrium in the solution phase. The retention time of the KTZ and CCFs is provided in Appendix A. The equilibrium solubilities of KTZ and its multi-components, as well as the solubilities of the corresponding CCFs, are presented in Table 4. The PXRD patterns of the residual solids are displayed in Appendix A. These results show that all the newly obtained solid forms exhibit significantly enhanced equilibrium concentrations compared with that of KTZ itself in pure water, the latter of which is almost insoluble. In particular, the greatest improvement in equilibrium solubility was observed in KTZ−GTA, and, accordingly, the solubility of GTA was also the largest among the five CCFs. On the one hand, the improved solubility of the KTZ multi-components could be ascribed to the higher aqueous solubility of the CCFs, since it has been previously proposed that cocrystal solubility is directly proportional to the solubility of its components [52,53]. On the other hand, reduced melting enthalpy of the multi-components would ultimately lead to high solubility [54,55]. KTZ−26DHB and KTZ−35DNB remained stable and nondissociable after the experiments and also showed the least solubility improvements among the five multi-components. It is interesting to note that solubility improvement of cocrystallization with VNA and PCA is more preferably achieved than by salt formation with 26DHB. These results suggest that the solubility of salts is not necessarily higher than that of cocrystals, since the solubility of multi-components is also related to factors such as energy, stability, and/or CCF solubility in some cases. The solubility of KTZ-PCA-I is approximately 500 μg mL^−1^ in ultrapure water [31], while the value is 386.34 ± 89.30 μg mL^−1^ in KTZ-PCA-II. The melting point of KTZ-PCA-I (142.9 °C) is lower than that of KTZ-PCA-II (151.5 °C), which corresponds to the trend in solubility values, i.e., a lower melting point with higher solubility.

### 2.8. Powder Dissolution Experiments

The solubility of KTZ exhibits strong pH dependence and is practically insoluble at higher pH values but is sparingly soluble under acidic conditions. The powder dissolution behaviors of KTZ and its five multi-components were investigated in ultrapure water and acetate buffer solution (pH 4.5), and the dissolution profiles are displayed in Figure 9. The results showed that the release of KTZ was faster, and the maximum solubility was significantly higher from all five multi-components than from KTZ alone in both media, which could mainly be attributed to the higher solubility of the CCFs and the dissociation behaviors of the multi-components. Consistent with equilibrium solubility, both KTZ−26DHB and KTZ−35DNB remained stable after the powder dissolution experiments, while the other three solid forms all dissociated into the KTZ free base (Appendix A). The trend of the dissolution profile for each of the multi-component crystals is the same in both media. The solubility of the CCF and solid-form stability in solution are two important parameters for dissolution, such that less stable and dissociable multi-components are reported to possess higher solubility than their stable counterparts [56]. Although the solubility of KTZ in pure water is significantly improved, the equilibrium solubility of KTZ in acidic media is not significantly improved. Whether the improvement of solubility and dissolution rate of the obtained multi-components leads to the increase in the oral bioavailability of KTZ needs to be further confirmed.

### 2.9. Stability

The stability results showed that throughout the testing period, the KTZ pure drug and the five obtained multi-component crystals were all stable at high temperature, high humidity, and light since there was no significant change in the PXRD patterns between the initial and final samples (Appendix A).

## 3. Materials and Methods

### 3.1. Materials

KTZ raw material (98%, C_26_H_28_Cl_2_N_4_O_4_) was purchased from Hubei Wande Chemical Co., Ltd. (Wande, China). The organic acids GTA (C_5_H_8_O_4_, 98%), VNA (C_8_H_8_O_4_, 98%), 26DHB (C_7_H_6_O_4_, 98%), PCA (C_7_H_6_O_4_, 98%), and 35DNB (C_7_H_4_N_2_O_6_, 98%) were purchased from Wuhan Yuancheng Co-create Technology Co., Ltd. (Wuhan, China). All the chemical reagents and analytical-grade solvents were purchased from the Beijing Chemical Reagent Factory (Beijing, China). All the chromatographical-grade solvents were purchased from Sigma Aldrich (St. Louis, MO, USA).

### 3.2. Sample Preparations

#### 3.2.1. Liquid-Assisted Grinding and Slurry

Liquid-assisted grinding and slurry methods were applied to prepare the KTZ multi-component crystals. Equimolar quantities of KTZ and the corresponding CCFs were weighed and added to a mortar, KTZ (1 mmol) with GTA (132.14 mg, 1 mmol), KTZ (531.43 mg, 1 mmol) with VNA (168.14 mg, 1 mmol), KTZ (531.43 mg, 1 mmol) with 26DHB (154.12 mg, 1 mmol), and KTZ (531.43 mg, 1 mmol) with 35DNB (212.12 mg, 1 mmol). The above complexes were ground using a pestle for 25–30 min, at which time several drops of ethyl acetate were added to the reactants. The resulting powder samples were submitted to PXRD for the identification of the phase formation. H. Zhang et al. [31] have reported a cocrystal form I of KTZ with PCA, hereafter referred to as KTZ−PCA−I, which is different from the form II (referred to as KTZ−PCA−II) found in this study. A physical mixture of KTZ (531.43 mg, 1 mmol) and PCA (154.12 mg, 1 mmol) was added into a glass vial, and then the suspension was left to be magnetically stirred for 6 h, at which time the solids were air-dried at room temperature to yield KTZ−PCA−II. The identity and purity of the above obtained solid forms were characterized and analyzed by PXRD and DSC.

#### 3.2.2. Slow Evaporation Method

Single crystal of KTZ−GTA 200 mg KTZ-GTA sample powders were dissolved in a 10 mL mixture of ethanol and acetone (1/1, *v*/*v*) solvent. The resulting solution was filtered and magnetically stirred for 12 h. Then, the filtrate was covered with Parafilm punctured with small holes and left at room temperature for solvent evaporation to occur. Approximately 4 days later, diffraction-quality block single crystals were harvested for SCXRD analysis.

Single crystal of KTZ−26DHB 100 mg KTZ−26DHB sample powders were dissolved in a 10 mL mixture of ethanol and acetone (1/1, *v*/*v*) solvent. After filtration, the resulting clear solution was allowed to evaporate slowly under ambient conditions. Colorless column crystals suitable for SCXRD analysis were precipitated out of the solution and isolated within 48 h.

Single crystal of KTZ−PCA−II 100 mg of KTZ−PCA sample powders were dissolved in 5 mL acetone solvent. After filtration and agitation, the resulting clear solution was allowed to evaporate slowly under ambient conditions. After 7 days, colorless column crystals were harvested and isolated for SCXRD analysis.

Single crystal of KTZ−35DNB 50 mg KTZ−35DNB sample powders were dissolved in a 10 mL mixture of ethanol and acetone (1/1, *v*/*v*) solvent. The resulting solution was filtered to give a clear solution and magnetically stirred for 12 h. Then, the filtrate was covered with Parafilm punctured with small holes and allowed to evaporate slowly under ambient conditions. Approximately 7 days later, yellow needle crystals were obtained for SCXRD analysis.

### 3.3. Single-Crystal X-ray Diffraction (SCXRD) Analysis

SCXRD measurements were conducted on a Rigaku Micromax 002+ diffractometer (The Woodlands, TX, USA) with Cu-*Kα* radiation (*λ* = 1.54184 Å) using a graphite monochromator at 295 K. The crystal structures were determined by SHELXT-2018 [57] via a direct method, and refinement was carried out by a full-matrix least-squares procedure against *F*^2^ [58] using the SHELXL-2018 program [59] with anisotropic displacement parameters for non-H atoms. All H atoms were placed in calculated positions and refined using a riding model. Graphical representations were prepared using the Mercury and Olex2 [60] programs. The crystallographic parameters of the four structures are summarized in Table 1. Information on the hydrogen-bonding interactions is provided in the Appendix A. The supplementary crystallographic data, including CIF for each refinement and checkcif files, have been deposited in the Cambridge Structural Database (CSD) (CCDC number 2253850–2253853) or can be retrieved from the supporting information.

### 3.4. Powder X-ray Diffraction (PXRD) Analysis

Bulk samples were analyzed on a Rigaku D/max−2550 diffractometer at 40 kV and 150 mA using Cu-*K*α radiation (*λ* = 1.54187 Å) (Rigaku, Tokyo, Japan) to obtain PXRD patterns. The measurements were performed at a continuous scan rate of 8° min^−1^ over a 2*θ* range of 3° to 40°. The data were further analyzed and imaged using Jade 6.5 software [61]. The simulated PXRD patterns were obtained using the Mercury program [62] based on the SCXRD data.

### 3.5. Thermal Analysis

DSC experiments were performed on a Mettler Toledo DSC-1 instrument (Mettler Toledo, Greifensee, Switzerland) [63]. Samples weighing approximately 5–6 mg were sealed in aluminum pans and subjected to heating from 30 to 250 °C at a heating rate of 10 °C min^−1^.

TGA was carried out on a Mettler Toledo TGA/DSC STARe calorimeter (Greifensee, Switzerland) [64]. Each sample (8–10 mg) was placed in an aluminum oxide pan and inserted into the TG furnace. The sample was heated with a heating rate of 10 °C min^−1^ in the range of 30–500 °C under a nitrogen purge gas at a flow rate of 50 mL min^−1^.

### 3.6. Fourier Transform Infrared Spectroscopy (FT−IR) Analysis

The FT−IR spectra [65] were collected using a PerkinElmer Spectrum 400 FT−IR instrument in the 650−4000 cm^−1^ range at 16 scans with a resolution of 4 cm^−1^.

### 3.7. Equilibrium Solubility

Equilibrium solubility assessments of KTZ and its multi-components were carried out using the shake flask method using a ZHWY−103D thermostatic shaker-incubator (Shanghai Zhicheng Analytical Instrument Manufacturing Co., Ltd., Shanghai, China) [66]. An excess amount of solid powder of pure KTZ and the prepared crystalline forms was added into glass vials containing 2 mL of ultrapure water and pH 4.5 acetate buffer solution, respectively. The solutions were stirred at 160 strokes min^−1^ with a magnetic bar at 37 °C for 48 h to reach equilibrium. The suspension was filtered through a 0.22 μm nylon filter membrane, diluted appropriately, and analyzed by subsequent high-performance liquid chromatography (HPLC) analysis. For each saturated solution, only the concentration of the KTZ was measured. An Agilent Eclipse XDB−C_18_ (250 mm × 4.6 mm, 5 μm) column was used throughout the process of elution with the column temperature set to 30 °C. The mobile phase, consisting of methanol and water (80/20, *v*/*v*), was run at 1.0 mL min^−1^ with a detection wavelength of 254 nm. The injection sample volume was 10 μL. The measurements were performed based on an external standard method. All the solubility experiments were performed in triplicate and the results are expressed as the mean value ± standard deviation (SD). Residual solids were collected after the equilibrium solubility experiment and characterized by PXRD for solid-form stability examinations.

### 3.8. Powder Dissolution Experiments

The powder dissolution experiments were performed via a dissolution apparatus (RC12AD dissolution tester, TDTF Technology Co., Ltd., Tianjin, China) by a basket method in pure water and pH 4.5 acetate buffer solution, respectively, with a rotation speed of 100 rpm under a temperature of 37 ± 0.2 °C [67]. KTZ and multi-component crystals weighing 200 mg KTZ powders (or corresponding to, for cocrystals) were added to a 600 mL dissolution medium. At regular time intervals of 5, 15, 30, 60, 90, 120, 180, 240, 360, and 480 min, 1 mL of solution was extracted, and the withdrawn solutions were filtered using a 0.45 μm syringe tip filter. The concentrations of KTZ were determined using the HPLC/UV method described above. All the dissolution experiments were performed in triplicate. After the dissolution experiments, residual solids were collected, air-dried, and confirmed by PXRD.

### 3.9. Stability Test

The accelerated stability tests of the KTZ and five multi-components were carried out in a drug stability test instrument (SHH−150SD) under accelerated storage conditions, high temperature (60 ± 1 °C), high humidity (90 ± 5%, 25 °C), and illumination (4500 ± 500 lx, 25 °C). Approximately 50 mg of powdered samples were stored under the three test conditions and measured at regular time intervals (0, 5, and 10 days) by PXRD [68].

## 4. Conclusions

In this paper, we synthesized five new KTZ derivatives, including four cocrystals with GTA, VNA, PCA, and 35DNB, and a salt containing a KTZ cation and a 26DHB anion. Our studies are in line with the trend of structural research and supramolecular synthon hierarchy in organic cocrystals that the −(C=O)−O−H. . .N-atom hydrogen bond is persistent in most cases, which implies the potential of organic acids in the field of cocrystal design for nitrogen heterocycle compounds. Comprehensive application of the p*K*_a_ rule, SCXRD, and crystallographic parameters were used to elucidate the ionization states. The equilibrium solubility and dissolution profiles both showed improvements compared with pure KTZ in two media after cocrystallization, which emphasizes the benefit of crystal engineering applied to improve the solubility and modify the release profile of poorly water-soluble drugs.

## Figures and Tables

**Figure 1 pharmaceuticals-16-01349-f001:**
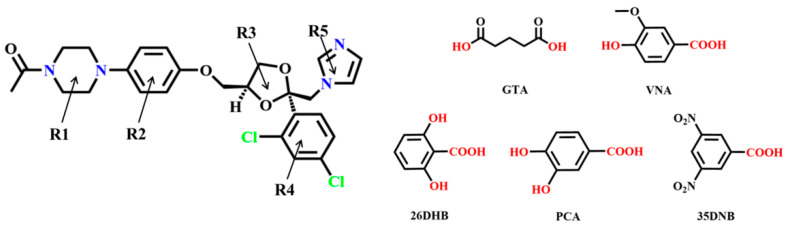
Chemical structure of KTZ and the organic acid coformers used in this study.

**Figure 2 pharmaceuticals-16-01349-f002:**
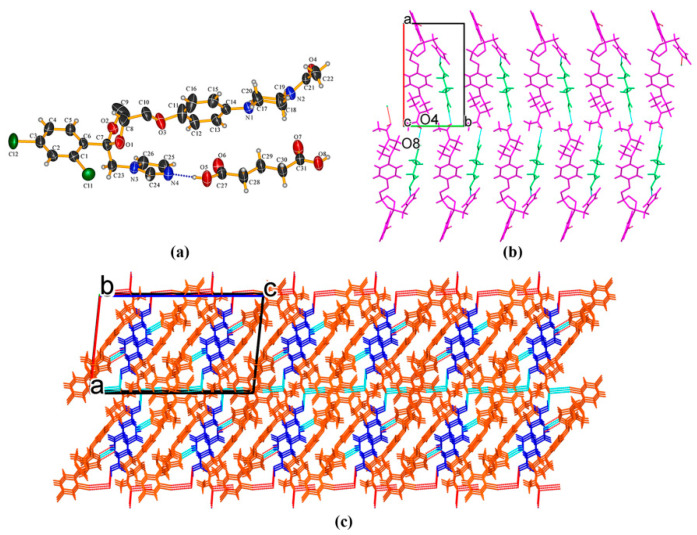
(**a**) Hydrogen-bonding interactions in an asymmetric unit; (**b**) KTZ molecules and GTA molecules appear alternately and stack along the crystallographic b-axis to form molecular chains; (**c**) crystal packing diagram viewed along the crystallographic b-axis.

**Figure 3 pharmaceuticals-16-01349-f003:**
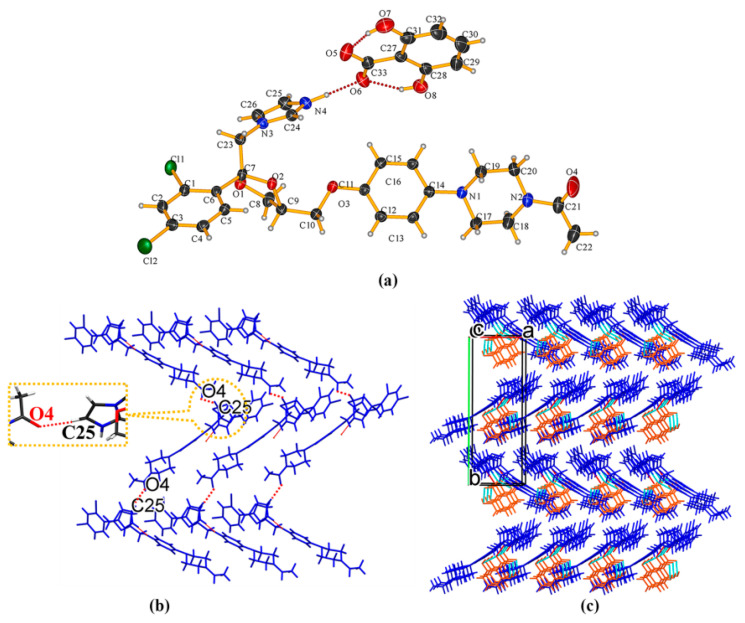
(**a**) Hydrogen-bonding interactions in an asymmetric unit; (**b**) C_25_−H_25_···O_4_ hydrogen-bonding interactions between adjacent two KTZ molecules; (**c**) “zig-zag” hydrogen-bonded chains stabilized by hydrogen bonds and other weak contacts.

**Figure 4 pharmaceuticals-16-01349-f004:**
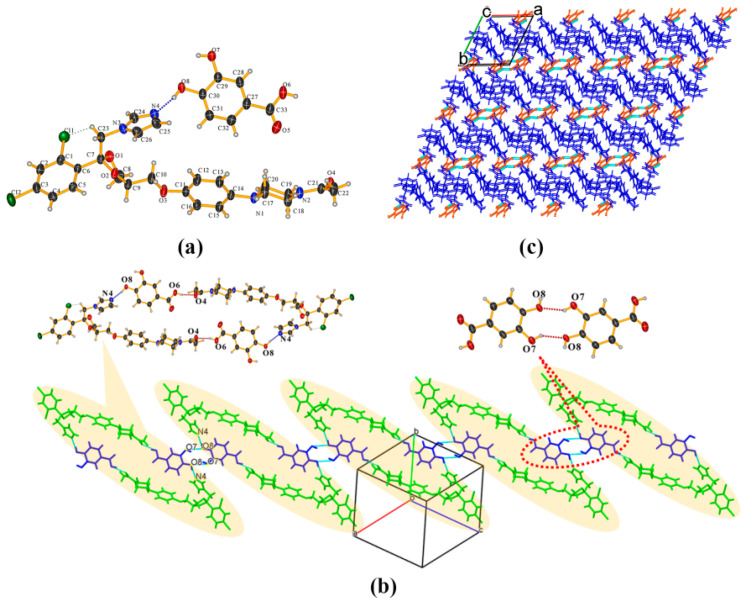
(**a**) Hydrogen-bonding interactions in an asymmetric unit; (**b**) annular tetrameric unit composed of two KTZ and two PCA molecules and homosynthon between adjacent two PCA molecules; (**c**) crystal packing diagram viewed along the crystallographic c-axis.

**Figure 5 pharmaceuticals-16-01349-f005:**
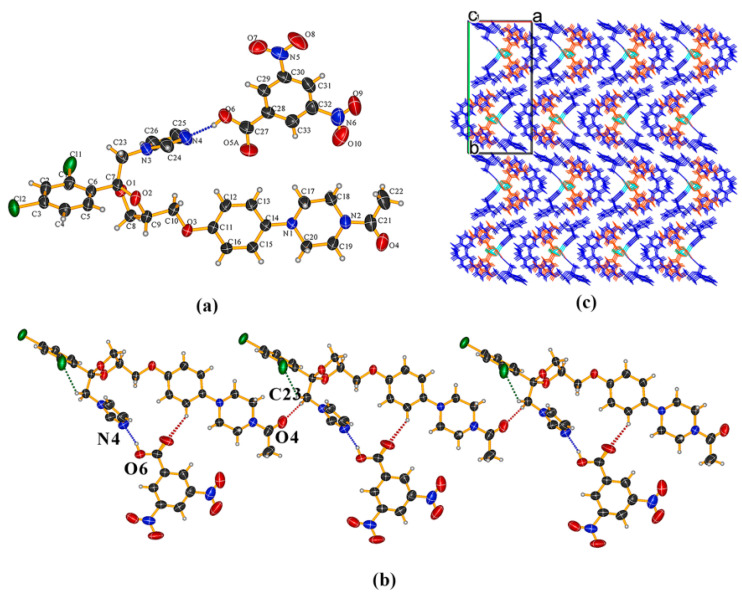
(**a**) Hydrogen-bonding interactions in an asymmetric unit; (**b**) adjacent asymmetric units connected via C_23_−H_23_···O_4_ hydrogen bonds between two KTZ molecules; (**c**) crystal packing diagram viewed along the crystallographic c-axis.

**Figure 6 pharmaceuticals-16-01349-f006:**
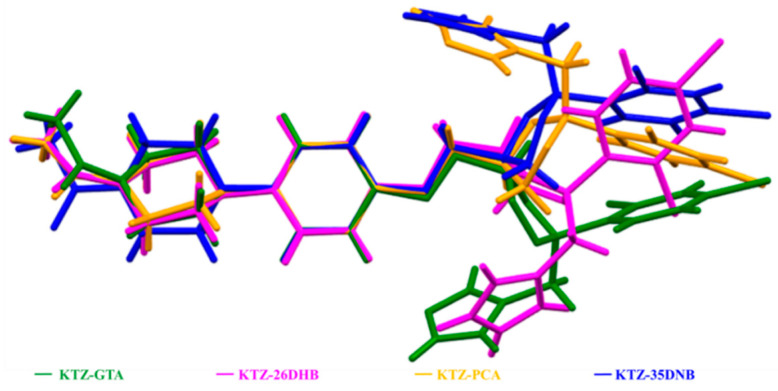
The conformation overlay of KTZ molecule in the four newly obtained multi-components.

**Figure 7 pharmaceuticals-16-01349-f007:**
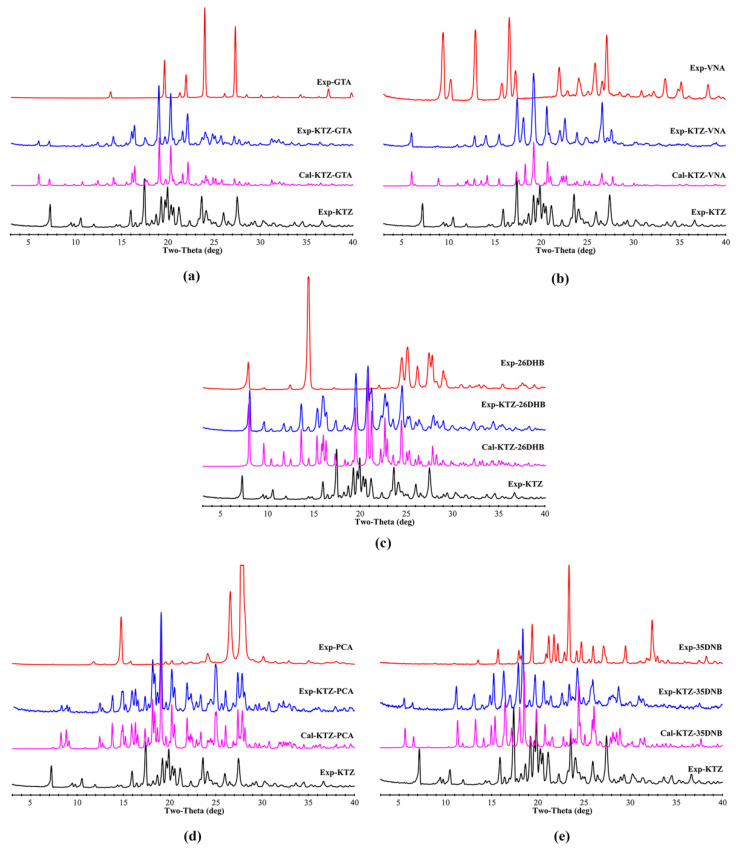
Comparative experimental and simulated PXRD patterns of the raw materials and multi-component crystals. (**a**) KTZ−GTA, (**b**) KTZ−VNA, (**c**) KTZ−26DHB, (**d**) KTZ−PCA-II, (**e**) KTZ−35DNB.

**Figure 8 pharmaceuticals-16-01349-f008:**
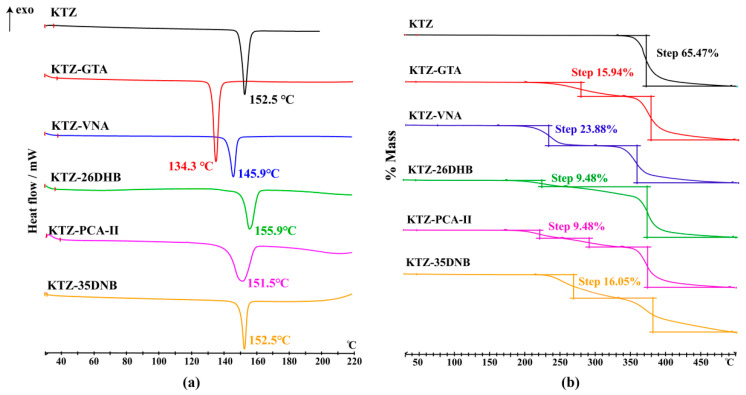
The (**a**) DSC and (**b**) TGA scans of KTZ and its multi-component crystals.

**Figure 9 pharmaceuticals-16-01349-f009:**
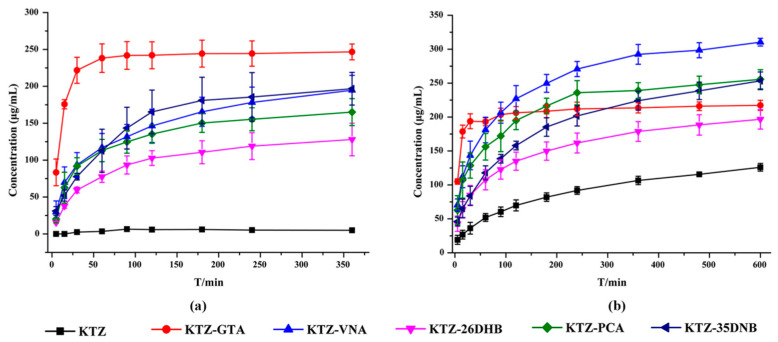
Powder dissolution profiles of KTZ and five multi-component crystals in (**a**) distilled water and (**b**) pH = 4.5 acetate buffer.

**Table 1 pharmaceuticals-16-01349-t001:** Crystallographic data and structure refinement details.

Title 1	KTZ−GTA	KTZ−26DHB	KTZ−PCA-II	KTZ−35DNB
Formula	C_26_H_28_Cl_2_N_4_O_4_, C_5_H_8_O_4_	(C_26_H_29_Cl_2_N_4_O_4_)^+^, (C_7_H_5_O_4_)^−^	C_26_H_28_Cl_2_N_4_O_4_, C_7_H_6_O_4_	C_26_H_28_Cl_2_N_4_O_4_, C_7_H_4_N_2_O_6_
Formula weight	663.54	685.54	685.54	743.54
Crystal size (mm)	0.15 × 0.17 × 0.23	0.28 × 0.33 × 0.44	0.21 × 0.28 × 0.44	0.11 × 0.14 × 0.46
Description	block	column	column	needle
Crystal system	monoclinic	monoclinic	triclinic	monoclinic
Space group	*P*2_1_/*c*	*P*2_1_	*P* − 1	*P*2_1_/*c*
Unit cell parameters (Å, °)	14.894 (1)	8.086 (1)	11.916 (1)	14.942 (1)
	8.757 (1)	22.186 (1)	12.090 (1)	31.193 (1)
	24.930 (1)	9.312 (1)	13.515 (1)	7.377 (1)
	90	90	99.35 (1)	90
	96.02 (1)	95.77 (1)	110.56 (1)	94.83 (1)
	90	90	110.41 (1)	90
Volume (Å^3^)	3233.74 (14)	1662.07 (11)	1615.97 (12)	3426.00 (6)
*Z*	4	2	2	4
Density (g cm^−3^)	1.363	1.370	1.409	1.442
Independent reflections	6542	6261	6152	6604
Reflections with *I* > 2*σ*(*I*)	5303	6094	5716	5748
*R_int_*	0.0571	0.0427	0.0417	0.0552
Final *R*, *wR*(*F*^2^) value	0.065, 0.171	0.040, 0.109	0.047, 0.128	0.044, 0.118
GOF	1.074	1.063	1.037	1.038
CCDC	2253852	2253850	2253853	2253851

**Table 2 pharmaceuticals-16-01349-t002:** p*K*_a_ values of CCFs, Δp*K*_a_ values between CCF and KTZ, C−O bond lengths of carboxylic/carboxylate group, and C_24_−N_4_−C_25_ bond angle.

Samples	p*K*_a_ Values	Δp*K*_a_ Values	C−O Bond Length (Å)	C=O(C−O^−^) Bond Length (Å)	C−N−C Bond Angle (°)
KTZ	6.5	-	-	-	104.1
KTZ−GTA	4.3	2.2	1.326/1.317	1.203/1.197	105.4
KTZ−26DHB	1.6	3.9	1.252	1.280	109.3
KTZ−PCA-II	4.5	2.0	1.318	1.205	105.6
KTZ−35DNB	2.8	3.7	1.285	1.235	106.6

**Table 3 pharmaceuticals-16-01349-t003:** The −C=O stretching vibration bands (cm^−1^) in CCFs and corresponding multi-component crystals.

Samples	−C=O Stretching	Samples	−C=O Stretching
KTZ	1644	/	/
GTA	1686	KTZ−GTA	1706
VNA	1673	KTZ−VNA	1698
26DHB	1663	KTZ−26DHB	1645
PCA	1667	KTZ−PCA-II	1693
35DNB	1698	KTZ−35DNB	1719

**Table 4 pharmaceuticals-16-01349-t004:** Equilibrium solubility of KTZ and multi-component crystals at ultrapure water and pH 4.5 buffer solution at 37.0 °C.

Samples	Solubility of CCFs (μg mL^−1^)	Concentration (μg mL^−1^)	
		Ultrapure Water	pH 4.5 Acetate Buffer
KTZ	-	1.20 ± 0.27	239.48 ± 4.69
KTZ−GTA	6500	2165.56 ± 251.34	339.87 ± 28.33
KTZ−VNA	4300	321.57 ± 10.03	365.44 ± 51.10
KTZ−26DHB	1600	139.13 ± 14.25	263.76 ± 8.51
KTZ−PCA	4500	386.34 ± 89.30	308.06 ± 33.95
KTZ−35DNB	2800	191.67 ± 15.95	273.66 ± 9.87

## Data Availability

CCDC 2253850–2253853 contains the supplementary crystallographic data for this paper. The data can be obtained free of charge via http://www.ccdc.cam.ac.uk/conts/retrieving.html (accessed on 1 September 2022) or by emailing data_request@ccdc.cam.ac.uk or by contacting The Cambridge Crystallographic Data Centre, 12 Union Road, Cambridge CB2 1EZ, UK; Fax: +44-1223-336033.

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
