# Peer review of "Enhancing Solubility and Dissolution Rate of Antifungal Drug Ketoconazole through Crystal Engineering"

_pharmaceuticals, 2023, doi:10.3390/ph16101349_

Round 1

Reviewer 1 Report

Hi,

The supplementary data is missing. The data needs to be uploaded because it includes some essential figures that the paper can not be accepted without checking them.

Kind Regards,

Author Response

Response to Reviewer 1 Comments

Point 1: The supplementary data is missing. The data needs to be uploaded because it includes some essential figures that the paper can not be accepted without checking them. 

Response 1: Thanks for your advice.

The supplementary data has been provided in the revised metarials.

Reviewer 2 Report

The authors enhanced the solubility and dissolution rate of an antifungal drug ketoconazole using crystal engineering technology. Five different co-crystals of ketoconazole were prepared and characterized physicochemically using various techniques. The prepared crystals were then evaluated for the solubility and dissolution studies. Overall, the work is interesting and will be beneficial for the fellow researchers. However, it requires major revision before its publication. My comments are as follows:

Abstract: The quantitative results are completely missing in the abstract. Authors are advised to include some quantitative results for the better understanding of the article.

Introduction: The introduction is quite shallow. The solubility part is too poor. Authors are advised to add recent literature about the importance of solubility and different solubility approaches. You can consult the following articles to make this manuscript more useful to the readers:

Molecules 25: E1559 (2020); J. Mol. Liq. 324: E115146 (2021); J. Mol. Liq. 331: E115700 (2021); J. Mol. Liq. 340: E117268 (2021).

Methods: Authors are advised include suitable literature for each experimental method.

Results and discussion: Please compare your results with previous studies and mention clearly how your work is important in comparison to already been reported.

Authors are advised to include the main limitation of work at the end of results and discussion section and just before the conclusion.

Avoid abbreviations before giving their explanation in text, figure, and table.

Conclusion: The conclusion should be concise and to the point indicating the application of the work.

The English language is fine. Minor editing is required.

Author Response

Response to Reviewer 2 Comments

The authors enhanced the solubility and dissolution rate of an antifungal drug ketoconazole using crystal engineering technology. Five different co-crystals of ketoconazole were prepared and characterized physicochemically using various techniques. The prepared crystals were then evaluated for the solubility and dissolution studies. Overall, the work is interesting and will be beneficial for the fellow researchers. However, it requires major revision before its publication. My comments are as follows:

Point 1: Abstract: The quantitative results are completely missing in the abstract. Authors are advised to include some quantitative results for the better understanding of the article. 

Response 1: Thanks for your advice.

The quantitative results have been added in the abstract in the revised manuscript.

The solubility of KTZ in distilled water significantly increased from 1.2 to 2165.6, 321.6, 139.1, 386.3, and 191.7 μg mL−1 in the synthesized multi-component forms with glutaric, vanillic, 2,6-dihydroxybenzoic, protocatechuic, 3,5-dinitrobenzoic acid, respectively. Especially for the cocrystal form with glutaric acid which has seen an 1800-fold solubility increase in water concerning KTZ. 

Point 2: Introduction: The introduction is quite shallow. The solubility part is too poor. Authors are advised to add recent literature about the importance of solubility and different solubility approaches. You can consult the following articles to make this manuscript more useful to the readers:

Molecules 25: E1559 (2020); J. Mol. Liq. 324: E115146 (2021); J. Mol. Liq. 331: E115700 (2021); J. Mol. Liq. 340: E117268 (2021).

Response 2: Thanks for your advice.

We have consulted and added above literature in the revised manuscript.

Point 3: Methods: Authors are advised to include suitable literature for each experimental method.

Response 3: Thanks for your advice.

Literature of for each experimental method has been added for each experimental method in the revised manuscript.

Point 4: Results and discussion: Please compare your results with previous studies and mention clearly how your work is important in comparison to already been reported.

Response 4: Thanks for your advice.

We have compared the results with previous studies in the results and discussion section in the revised manuscript. There are three points worth emphasizing in this study: (i) It was found that salts are not necessarily more soluble than cocrystals and that correlations between salt/cocrystal solubility, aqueous stability, or coformer solubility are compared. Solubility of the CCF and solid-form stability in solution are two important parameters for dissolution, such that less stable and dissociable multi-components are reported to possess higher solubility than their stable counterparts. The solubility values of cocrystal polymorph correspond to the trend of melting points; (ii) pKa rule, SCXRD and crystallographic parameters were used complementarily to elucidate the proton transfer, thus judging the formation of salt or cocrystal accurately; (iii) By analyzing conformation of KTZ, it was found that the introduction of CCF molecules into the crystal lattice or hydrogen bonds between API and CCF enables the torsion of API conformation, at the same time, the API molecule with flexible conformation forms a more matching spatial arrangement between API and CCF, which promotes the formation of cocrystals.

Point 5: Authors are advised to include the main limitation of work at the end of results and discussion section and just before the conclusion.

Response 5: Thanks for your advice.

The main limitation of work has been added in the revised manuscript. There are mainly two limitations of this work: (i) Although the solubility of KTZ in pure water is significantly improved, the equilibrium solubility of KTZ in acidic media is not significantly improved; (ii) Whether the improvement of solubility and dissolution rate of the obtained multi-components leads to the increase the oral bioavailability of KTZ needs to be further confirmed.

Point 6: Avoid abbreviations before giving their explanation in text, figure, and table.

Response 6: Thanks for your advice.

Abbreviations of “API” have been deleted before giving their explanation in text, figure, and table.

Point 7: Conclusion: The conclusion should be concise and to the point indicating the application of the work.

Response 7: Thanks for your advice.

The conclusion has been improved to be more concise and to the point indicating the application of the work as follows: 

In this paper, we synthesized five new KTZ derivatives, including four cocrystals with GTA, VNA, PCA, and 35DNB, and a salt containing a KTZ cation and a 26DHB anion. Our studies are in line with the trend of structural research and supramolecular synthon hierarchy in organic cocrystals that −(C=O)−O−H. . .N-atom hydrogen bond is persistent in most cases, which implies the potential of organic acids in the field of cocrystal design for nitrogen heterocycle compounds. Comprehensive application of pKa rule, SCXRD and crystallographic parameters were used to elucidate the ionization states. The equilibrium solubility and dissolution profiles both showed improvements compared with pure KTZ in two media after cocrystallization, which emphasizes the benefit of crystal engineering applied to improve the solubility and modify the release profile of poorly water-soluble drugs.

Reviewer 3 Report

The manuscript describes preparing and characterizing various multicomponent solid forms containing ketoconazole (cocrystals, salt, polymorph). It is worth noting that they used different synthesis methods: liquid-assisted grinding (VALAG), slurry, and slow evaporative procedures.

1)      To obtain the KTZ-PCA II polymorph in this solid form, did you try the synthesis with LAG but using different solvents and varying amounts? It is mentioned because it has been observed that obtaining a polymorph can be favored using VALAG (variable amount liquid-assisted grinding).1,2 2)      In 2.2.1., the authors mention that Zhang et al. reported the polymorph KTZ-PCA I, but the citation is not given at this point. Subsequently, in 3.7, the solubility of this same polymorph is mentioned and refers to the citation [27], and until that moment, the citation is known. 3)      In point 3.2, about the KTZ-26DHB salt, to differentiate between the cocrystal, the salt, and the salt-cocrystal continuum, to know the degree of ionization and proton transfer of the species, 15N solid-state NMR and XPS are used.3,4 These techniques have been previously employed in solid forms containing KTZ to differentiate this.5,6 In these works, they used 15N ssNMR. In this way, it is recommended that the authors carry out this analysis to say conclusively if it is a salt or a cocrystal. Authors are requested to attend to these major corrections to make the manuscript suitable for publication in Pharmaceuticals.  

(1)      Hasa, D.; Miniussi, E.; Jones, W. Mechanochemical Synthesis of Multicomponent Crystals: One Liquid for One Polymorph? A Myth to Dispel. Cryst. Growth Des. 2016, 16 (16), 4582–4588. https://doi.org/https://doi.org/10.1021/acs.cgd.6b00682.

(2)      Solares-Briones, M.; Coyote-Dotor, G.; Páez-Franco, J. C.; Zermeño-Ortega, M. R.; de la O Contreras, Ca. M.; Canseco-González, D.; Avila-Sorrosa, A.; Morales-Morales, D.; Germán-Acacio, J. M. Mechanochemistry : A Green Approach in the Preparation of Pharmaceutical Cocrystals. Pharmaceutics 2021, 13, 790. https://doi.org/10.3390/pharmaceutics13060790.

(3)      Tothadi, S.; Shaikh, T. R.; Gupta, S.; Dandela, R.; Vinod, C. P.; Nangia, A. K. Can We Identify the Salt–Cocrystal Continuum State Using XPS? Cryst. Growth Des. 2021, 21, 735–747. https://doi.org/10.1021/acs.cgd.0c00661.

(4)      Stevens, J. S.; Byard, S. J.; Seaton, C. C.; Sadiq, G.; Davey, R. J.; Schroeder, S. L. M. Proton Transfer and Hydrogen Bonding in the Organic Solid State: A Combined XRD/XPS/SsNMR Study of 17 Organic Acid-Base Complexes. Phys. Chem. Chem. Phys. 2014, 16 (3), 1150–1160. https://doi.org/10.1039/c3cp53907e.

(5)      Martin, F. A.; Pop, M. M.; Borodi, G.; Filip, X.; Kacso, I. Ketoconazole Salt and Co-Crystals with Enhanced Aqueous Solubility. Cryst. Growth Des. 2013, 13 (10), 4295–4304. https://doi.org/10.1021/cg400638g.

(6)      Chen, X.; Li, D.; Deng, Z.; Zhang, H. Ketoconazole: Solving the Poor Solubility via Cocrystal Formation with Phenolic Acids. Cryst. Growth Des. 2020, 20 (10), 6973–6982. https://doi.org/10.1021/acs.cgd.0c01014.

Author Response

Response to Reviewer 3 Comments

Point 1: The manuscript describes preparing and characterizing various multicomponent solid forms containing ketoconazole (cocrystals, salt, polymorph). It is worth noting that they used different synthesis methods: liquid-assisted grinding (VALAG), slurry, and slow evaporative procedures. To obtain the KTZ-PCA II polymorph in this solid form, did you try the synthesis with LAG but using different solvents and varying amounts? It is mentioned because it has been observed that obtaining a polymorph can be favored using VALAG (variable amount liquid-assisted grinding).1,2

(1)      Hasa, D.; Miniussi, E.; Jones, W. Mechanochemical Synthesis of Multicomponent Crystals: One Liquid for One Polymorph? A Myth to Dispel. Cryst. Growth Des. 2016, 16 (16), 4582–4588. https://doi.org/https://doi.org/10.1021/acs.cgd.6b00682.

(2)      Solares-Briones, M.; Coyote-Dotor, G.; Páez-Franco, J. C.; Zermeño-Ortega, M. R.; de la O Contreras, Ca. M.; Canseco-González, D.; Avila-Sorrosa, A.; Morales-Morales, D.; Germán-Acacio, J. M. Mechanochemistry : A Green Approach in the Preparation of Pharmaceutical Cocrystals. Pharmaceutics 2021, 13, 790. https://doi.org/10.3390/pharmaceutics13060790.

Response 1: Thanks for your advice.

Five solvents of methanol, ethanol, ethyl acetate, acetone and acetonitrile were used to prepare KTZ-PCA, and the doses varied from 200uL, 500uL, 1mL, 2mL to 5mL, when a physical mixture of KTZ (531.43 mg, 1 mmol) and PCA (154.12 mg, 1 mmol) was added into a glass vial. It turned out that the final polymorph was independent of the solvent type and dose, but only related to the preparation method. The LAG method resulted in KTZ-PCA-I, while the slurry method led to KTZ-PCA-II.

Point 2: In 2.2.1., the authors mention that Zhang et al. reported the polymorph KTZ-PCA I, but the citation is not given at this point. Subsequently, in 3.7, the solubility of this same polymorph is mentioned and refers to the citation [27], and until that moment, the citation is known. 

Response 2: Thanks for your advice.

The related citation has been added in the section of 2.2.1 after mentioning Zhang et al when it is mentioned firstly.

Point 3: In point 3.2, about the KTZ-26DHB salt, to differentiate between the cocrystal, the salt, and the salt-cocrystal continuum, to know the degree of ionization and proton transfer of the species, 15N solid-state NMR and XPS are used.3,4 These techniques have been previously employed in solid forms containing KTZ to differentiate this.5,6 In these works, they used 15N ssNMR. In this way, it is recommended that the authors carry out this analysis to say conclusively if it is a salt or a cocrystal. Authors are requested to attend to these major corrections to make the manuscript suitable for publication in Pharmaceuticals.  

(3)      Tothadi, S.; Shaikh, T. R.; Gupta, S.; Dandela, R.; Vinod, C. P.; Nangia, A. K. Can We Identify the Salt–Cocrystal Continuum State Using XPS? Cryst. Growth Des. 2021, 21, 735–747. https://doi.org/10.1021/acs.cgd.0c00661.

  • Stevens, J. S.; Byard, S. J.; Seaton, C. C.; Sadiq, G.; Davey, R. J.; Schroeder, S. L. M. Proton Transfer and Hydrogen Bonding in the Organic Solid State: A Combined XRD/XPS/SsNMR Study of 17 Organic Acid-Base Complexes. Phys. Chem. Chem. Phys. 2014, 16 (3), 1150–1160. https://doi.org/10.1039/c3cp53907e.

(5)      Martin, F. A.; Pop, M. M.; Borodi, G.; Filip, X.; Kacso, I. Ketoconazole Salt and Co-Crystals with Enhanced Aqueous Solubility. Cryst. Growth Des. 2013, 13 (10), 4295–4304. https://doi.org/10.1021/cg400638g.

(6)      Chen, X.; Li, D.; Deng, Z.; Zhang, H. Ketoconazole: Solving the Poor Solubility via Cocrystal Formation with Phenolic Acids. Cryst. Growth Des. 2020, 20 (10), 6973–6982. https://doi.org/10.1021/acs.cgd.0c01014. 

Response 3: Thanks for your advice.

From the literature, both of 15N solid-state nuclear magnetic resonance (NMR) and XPS can be used to differentiate cocrystal from salt and the salt-cocrystal continuum. But the instrumentations are not widely available. In this paper, we have used two commonly used methods (ΔpKa rule and SCXRD) to distinguish between cocrystal, salt, and salt-cocrystal continuum. The bond length difference between two C−O bonds in the carboxylate group ΔdC−O is useful to determine the proton transfer. The small ΔdC−O value indicates the  proton transfer and the formation of a salt, while the large value suggests the formation of cocrystal. The results provided evidence for the salt/cocrystal formation [1-4]. The residual density map provided by the SXRD data can also be used to distinguish the location of the H atom and thus the cocrystal, salt, and salt-cocrystal continuum could be distinguished. In the KTZ−GTA, KTZ−PCA-II, and KTZ−35DNB, the H atom was close to the oxygen atom on the carboxylic acid group which can be regarded as a cocrystal, while in the KTZ−26DHB, the H atom was close to the nitrogen atom on the imidazole ring, which means the KTZ−26DHB should be regarded as a salt.

KTZ−GTA         KTZ−PCA-II        KTZ−35DNB           KTZ−26DHB

(only the imidazole ring of the KTZ molecule showed to indicate the H atom position in the KTZ−GTA, KTZ−PCA-II, KTZ−35DNB, and KTZ−26DHB)

[1] Wu, D., Li, J., Xiao, Y., et al. New salts and cocrystals of pymetrozine with improvements on solubility and humidity stability: Experimental and theoretical study[J]. Crystal Growth & Design, 2021, 21(4): 2371-2388.

[2] Zhao, Y., Sun, B., Jia, L., et al. Tuning physicochemical properties of antipsychotic drug aripiprazole with multicomponent crystal strategy based on structure and property relationship[J]. Crystal Growth & Design, 2020, 20(6): 3747-3761.

[3] Chen, J. M., Wang, Z. Z., Wu, C. B., et al. Crystal engineering approach to improve the solubility of mebendazole[J]. CrystEngComm, 2012, 14(19): 6221-6229.

[4] Lehtonen, O., Hartikainen, J., Rissanen, K., et al. Hydrogen bonding and protonation in acid–base complexes: Methanesulfonic acid-pyridine[J]. The Journal of Chemical Physics, 2002, 116(6): 2417-2424.

Round 2

Reviewer 1 Report

Accept in present form

Reviewer 2 Report

The authors have addressed the previous concerns. The revised manuscript is suitable for publication in its present form.

Reviewer 3 Report

Previous questions have been answered satisfactorily. I consider the manuscript ready to be published in Pharmaceuticals.